# Water Asset Transition through Treating Water as a New Asset Class for Paradigm Shift for Climate–Water Resilience

Amgad Elmahdi * and Lixiang Wang

Green Climate Fund—GCF, Incheon 22004, Republic of Korea
* Correspondence: aelmahdi@gcfund.org; Tel.: +82-1074586239

**Abstract:** Climate change is evident around the globe, which requires bold actions now to achieve UN-SDGs and Paris Agreement. The water sector is dominated by public finance and is almost subsidised. In addition, there is an increased risk perception surrounding climate investments in developing countries. Pricing climate risks is a daunting challenge for investors and the private sector, who must estimate the likelihood of various climate scenarios and their implications for physical, liability and transition risks at the firm, project, national, and regional scales. In addition, there is a building momentum to scale up global climate response. To translate this momentum into action will require significantly greater investments, investments in a different set of inclusive assets that address water security, mobilise the private sector and provides sector-based or economy-wide co-benefits to direct and indirect beneficiaries, e.g., job creation, health benefits, improved resilience and scaling knowledge and harmonise data and methodologies. Notably, climate–water finance is facing a dual challenge. It will have to both reduce the present water infrastructure financing gap and ensure that this new infrastructure/asset is low-carbon, resilient to climate change, and meets the goals of the UNFCCC and the Paris Agreement. Therefore, there is a need for a paradigm shift in the way how water asset is defined, developed, and financed. This paper presents this novel approach concept and its content and financial structure that enable treating water as a new asset class to enable private sector investment and ensure providing water for domestic, municipal, and industrial purposes and allows municipalities to scale their water reuse, sanitation, and desalination projects in partnership with the private sector and/or governments. It is increasingly important to treat water as a new asset class, particularly as nations around the world (particularly developing countries) are set to experience an anticipated 40% shortfall in water by 2030 due to climate change, economic recovery and growth, population growth and resource competition. Investment in water could be one of the ways of tackling this deficit by treating water as a new asset class.

**Keywords:** water asset; asset class; water reuse; sanitation

## 1. Introduction

Climate change is evident around the globe, which requires bold actions now to achieve UN-SDGs and Paris Agreement. The projected impacts of climate change will reduce accessibility to water resources and escalate spatial and seasonal variations in water availability. Growing uncertainty and variability in climate, particularly precipitation, means increasing intensity and frequency of drought and flood, which in turn impact the water infrastructure/assets including wastewater and sanitation. The socioeconomic costs of floods and droughts—for which wastewater assets and storages are key mitigation and adaptation measures—are growing.

In addition, water is one of the world's most essential commodities and as the global population continues expanding, water supply becomes one of the most important issues and demand for water services is growing in many places due to economic growth. Facing the increasing demand for this scarce resource, the global water business seems to offer investment opportunities. The recent analysis provides a partial estimate of the scale of

global economic losses related to water security: USD 260 billion per year from inadequate water supply and sanitation, USD 120 billion per year from urban property flood damages, and USD 94 billion per year to existing irrigators [1].

Water infrastructure systems include the entire water resources, water supply and sanitation chains, including water extraction and transmission, water treatment and supply, and wastewater collection, treatment, and disposal. Those systems are most likely compromised under climate change scenarios due to variable water availability from changes in rainfall patterns and glacial melting and increases in extreme weather events causing drought and flooding. The vision for a paradigm shift in water security for water infrastructure systems needs to demonstrate how water resources are managed in an integrated manner, water supply is treated and distributed, and wastewater is collected, treated, and disposed of. To achieve the vision, appropriate finance and tariffs and fees support an equitable water distribution and provide adequate financing for O&M and future investment.

There is a building momentum to scale up global climate response. To translate this momentum into action will require significantly greater investments, investments in a different set of inclusive assets that address water security and provide sector-based or economy-wide co-benefits to direct and indirect beneficiaries, e.g., job creation, health benefits, and improved resilience. Notably, climate finance is facing a dual challenge. It will have to both reduce the present water infrastructure financing gap and ensure that this new infrastructure/asset is low-carbon, resilient to climate change, and meets the goals of the UNFCCC and the Paris Agreement. According to some estimates, the infrastructure gap worldwide could reach USD 3 to USD 15 trillion by 2040 [2]. Therefore, there is a need for a paradigm shift in the way how water asset is defined, developed, and financed. It is increasingly important to treat water as a new asset class, particularly as nations around the world (particularly developing countries) are set to experience an anticipated 40% shortfall in water by 2030 due to climate change, economic recovery and growth, population growth and resource competition.

Investment in water could be one of the ways of tackling this deficit by treating water as a new asset class. As quoted by World Bank, "Water is the best investment the world can make to improve health, food security, gender equality, and the environment while transforming lives & communities" [3]. In the context of climate adaptation, the GCF-Green Climate Fund is exploring non-conventional water sources—including wastewater in general and water reuse and water recycling in particular—as a new asset class. In some regions, reused water (untapped resources continue to increase as population increase) is already a water source (almost 80% of wastewater-untapped resources is untreated); however, there are several barriers to broader adoption in many other countries including financial market barriers and assuring affordability and bankability of such projects.

### 1.1. Proposed Solution

The water sector needs to maximise the value of existing and new assets for water-related investments: service providers can reduce overall investment needs and improve capital efficiency through operational improvement. Efficiency often results from better O&M with the objective to improve the service delivery to the users. In addition, GCF's goal statement for water security [4] is "GCF promotes a paradigm shift in water security that is low-carbon, resilient to climate change, and meets the goals of the UNFCCC and Paris Agreement":

- IF the GCF creates an enabling investment environment to identify, design, and implement public and privately funded transformational water security interventions as a new asset class;
- THEN GCF's recipient countries can simultaneously mitigate and adapt to climate change through two low carbon climate-resilient development pathways in (i) water conservation; and (ii) preservation of water;

- BECAUSE an increasing share of investment in water security will be catalysed to deliver systemic change and maximise impact across the drivers of change in water security (Catalysing climate innovation and Mobilising finance at scale).

In simple means, this aims to create an enabling credit enhancement and blended financing environment through alternative funding solutions and the establishment of water reuse/sanitation infrastructure as a new water asset by defining the investment value of the asset and creating the enabling financial and institutional environment to take the asset to private market and investor.

### 1.2. Definition

The new asset class is "an asset for adaptation and/or mitigation that is developed and funded using credit enhancement to crowd in private sector funding targeted towards developing debt capital market and acceptable revenue streams but remains in line with ESG impacts and help to meet the targets set in the Paris Agreement and contribute to UN SDGs (Goal 6—clean water and sanitation; Goal 3—Good health and wellbeing, Goal 5—Gender Equality, Goal 7—Affordable and Clean Energy; Goal 13—Climate Action; Goal 14—Sustainable Oceans and Goal 17—Partnerships with the Involvement of the Private Sector) and providing water for domestic, municipal, and industrial purposes and allows municipalities to scale their water reuse projects in partnership with the private sector and/or governments purchase a service instead of an asset".

### 1.3. Financial Barriers and Solutions

The recent analysis provides a partial estimate of the scale of global economic losses related to water security and how much global financing needs for water infrastructure that range from USD 6.7 trillion by 2030 to USD 22.6 trillion by 2050. However, there are barriers creating a gap between current financing and future needs, including these barriers to financing water security [5–8].

- Under-valuing of water: Water is a public good and generally is under-valued and is not properly accounted for by the government and the investors that depend on or affect its availability in other sectors such as urban development, agriculture, and energy;
- Water services under-priced: Water services are often under-priced, resulting in low cost recovery for water investments;
- Capital-intensive: Water resources, irrigation, water supply, and wastewater infrastructures are generally capital-intensive, with high sunk costs and long payback periods;
- Difficulty of monetising benefits and co-benefits: Water management provides both public and private co-benefits, many of which cannot be easily monetised. This reduces potential revenue flows and the availability of creditworthiness;
- Context-specific projects: Water projects are often too small or too context-specific, raising transaction costs and making innovative financing models difficult to scale up;
- Poor business models: Business models often fail to support O&M efficiency, hampering the ability to sustain service at least cost over time in addition to integrity and transparency context.

These barriers are hindered, by investors and the private sector, from water projects. To overcome these barriers, the High-Level Panel on Water has defined a range of principles to help finance investments, enhance water services, mitigate water-related risks, and contribute to sustainable growth. In addition, one of the four-pronged GCF delivery mechanism is mobilising finance. To do that for the new asset, GCF's goal is IF finance is deployed to reduce risks and barriers of water security interventions, THEN financial resources will catalyse private and commercial finance at scale to support the paradigm-shifting pathways of water conservation and preservation of water BECAUSE the financial viability of new asset classes in water security will be demonstrated. The creation of a new

asset class through the development of blended finance and an effective 'take-it-to-market' approach would need to ensure:

- De-risking water security investments;
- Scaling up blended finance into water security interventions;
- Increasing collaboration with financial partners.

In this paper, we are proposing this innovative financial model (Figure 1) for the new asset class that will address financial market barriers and ensure affordability and bankability to unlock water reuse and sanitation investment, and how different financing options will overcome these barriers, including

- Reclassify and redistribute the responsibilities among the players across infrastructure design, procurement, construction (capital expenditure) and the long-term operations and maintenance (O&M) of the asset;
- Access to water is a human right; however, affordable tariffs remain a challenge in revenue models, for which investors seek full-cost recoverable tariffs from water reuse and sanitation projects;
- Constraints on commercial loan grace and maturation periods, which require such loans to be carefully structured to better suit investments in long-term infrastructure assets such as water reuse and sanitation projects;
- Public finance restrictions, particularly those which limit the extent to which municipalities can commit to long-term financing.

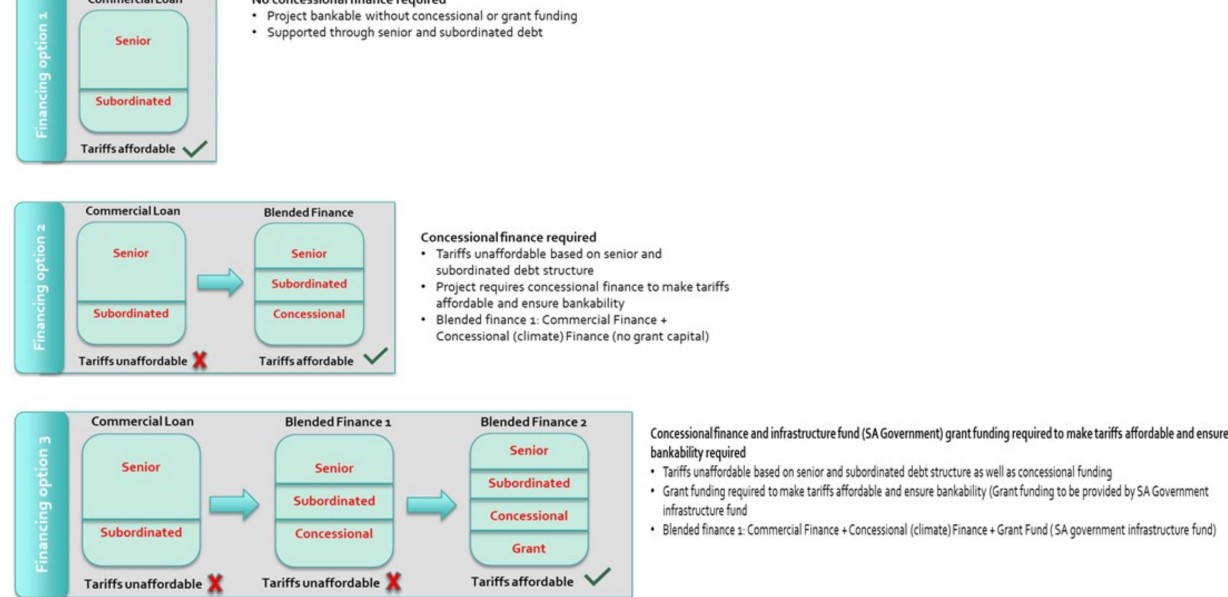

**Figure 1.** Financing options and affordability.

Treating water as "a new asset class" for water reuse and sanitation will allow municipalities and the private sector to scale up water reuse, sanitation, and desalination projects and/or governments to purchase a service instead of an asset. GCF's role will facilitate this approach using the concessional through:

- Supporting countries to develop and adopt policies and legislation to create an enabling investment environment to identify, design, and implement public and privately funded transformational water security interventions as a new asset class;
- Finance the transition and de-risk private investment to address financial market barriers and ensure affordability and bankability to unlock water reuse investment;
- Supporting new financial models accompanied with acceptable revenue in line with Paris agreement targets and SDGs.

## 2. New Asset Class-Main Characteristics

To ensure the delivery of the financial approach to the new asset and the creation of this new class, the following four elements (shown in Figure 2) are essential in the design, development and financing of the new asset class:

- Financial;
- SDGs;
- Paris Agreement targets;
- Acceptable revenue in line with SDGs, ESG impacts and Paris Agreement.

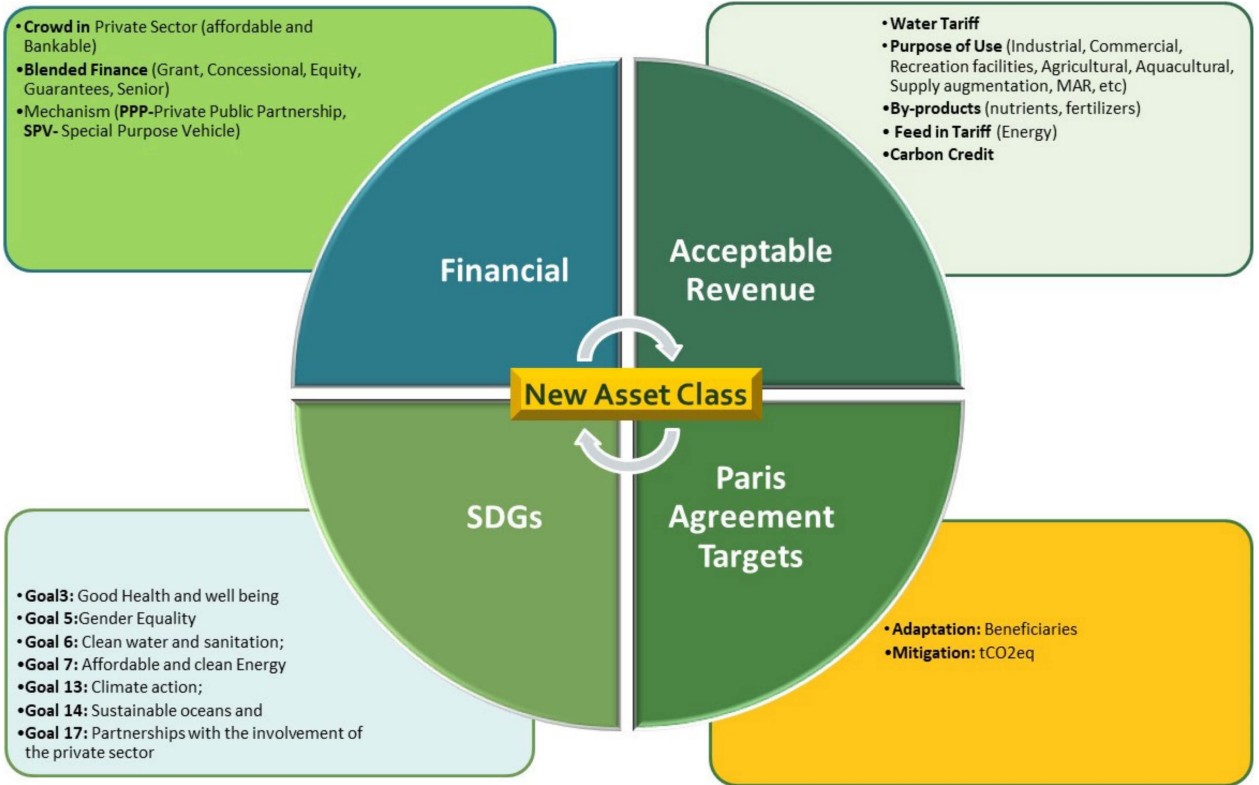

**Figure 2.** Key Characteristics of the New Asset Class.

Treating water as a new asset class for water security requires an enabling investment environment, water resource conservation, integrated approaches to water management, and regenerative acceptable revenue approaches to economic development, and planning for climate change resilience. However, it is acknowledged there are many barriers to achieving this paradigm shift for water security, most of which apply across the entire water sector. Key actions to overcome these barriers include:

- Cultural change: achieving water security involves creating a culture and environment that allows changes to take hold and work in practice;
- Collaboration: collaboration is essential for inspiring new ideas and applications, allowing for insights to develop, and spurring innovation. In addition to collaborating with external stakeholders, water sector actors can collaborate within their organisations, with other organisations, and with partners outside of the water sector at the scale (the water sector is the connector between many sectors, i.e., food, energy, ecosystems, etc.) [9];
- Technology: technology, when paired with the right culture, processes, and people, is a powerful enabler of innovation;
- Innovative regulatory frameworks: regulatory frameworks can be adequately designed to challenge various actors in the water sector to improve innovation for the benefit of customers, the environment, and broader society including inclusive

governance arrangements. Regulatory frameworks can encourage innovation by developing market-based instruments to recover the full cost of providing water and related services and encourage research and development in innovative projects.

These actions will drive changes and uptake of this innovative approach of treating water as a new asset class in parallel with establishing several enablers. These actions and enablers must address the underlying conditions that create or hinder transformation and the fundamental systems that support that transformation. It is also crucial to understand that, while the barriers are often assigned to separate systems, they are interconnected and as such enablers must address the interconnected nature of the barriers. These enablers are:

- Apply scientific evidence to support decision-making and develop a response to future needs;
- Investment programmes need to build the long-term capacity of local actors, rather than short-term delivery efficiency;
- Change how success is measured and make it culturally and locally specific and tailored;
- Embrace the central role of an IWRM approach to addressing water security challenges yet recognise that water is not the only actor (centre for SDGs);
- Innovative technical and social approaches need to bundle to create impact at a system scale;
- Technological adaptation and suite of innovations (technical, financial, business, institutional and social) to climate-related water security challenges in treating wastewater and re-using need to account for local context and access to markets and finance.

### 3. Conclusions and Key Messages

Increasing the resilience of the water sector to climate change requires a paradigm shift in the way how water asset is defined, developed, and financed. This involves several and bold actions now:

- Increase focus on adaptation that in turn supports a new asset class in sanitation and water reuse;
- Develop and apply a new financial model through credit enhancement that is accompanied by acceptable revenue streams that are in line with ESG impacts, Paris agreement targets and SDGs. This is essential for the recognition of wastewater and sanitation facilities as an asset class for private investment in developing countries;
- Building capacities of project owners in structuring bankable and affordable projects to encourage investors' economic interest in water and sanitation projects and take it to the market;
- Provision of innovative financing solutions including the use of credit enhancement and blended finance mechanisms that lowers the cost of borrowing and improve investment grade levels;
- Creating partnerships and strengthening investor relationships among governments, financial institutions, and wider stakeholders to improve investors' understanding and confidence;
- GCF's role in financing the transition can make it happen through supporting countries to develop and adopt policies and legislation to create an enabling investment environment; financing the transition and de-risk private investment in addressing financial market barriers and ensuring affordability and bankability and supporting new financial models accompanied by acceptable revenue in line with Paris agreement targets and SDGs.

**Author Contributions:** Conceptualization, methodology, writing—original draft preparation, visualization, investigation, A.E.; writing—review and editing, resources, project administration, L.W. All authors have read and agreed to the published version of the manuscript.

**Funding:** This paper received no external funding.

**Conflicts of Interest:** The authors declare no conflict of interest.

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
