# Peer review of "Water Asset Transition through Treating Water as a New Asset Class for Paradigm Shift for Climate–Water Resilience"

_climate, doi:10.3390/cli10120191_

Round 1

Reviewer 1 Report

This article tackled very important issues to introduce the asset class approach for water reuse and sanitation, and to promote the new asset class approach among the public and private water sector actors, investors, and wider community. However, the paper is not properly structured and the writing style is very difficult to read. The author should re-structure the text to make it more understandable for the readers. Here are some comments:

(1)   Section2 and 3. Please write with more awareness of paragraph writing.

(2)   Page 2. Please cite the literature on GCF-Green Climate Fund.

(3)   Fonts are different in many places making it very difficult to read.

(4)   Figures are unclear and illegible.

(5)   References should be arranged in alphabetical order and written in a consistent manner.

Author Response

Thanks for the reviewer and much appreciated your valuable comments to improve the paper

This article tackled very important issues to introduce the asset class approach for water reuse and sanitation, and to promote the new asset class approach among the public and private water sector actors, investors, and wider community. However, the paper is not properly structured and the writing style is very difficult to read. The author should re-structure the text to make it more understandable for the readers. Here are some comments: The paper restructured

(1)   Section2 and 3. Please write with more awareness of paragraph writing. Done

(2)   Page 2. Please cite the literature on GCF-Green Climate Fund.  Citation provided

(3)   Fonts are different in many places making it very difficult to read. Consistent font revisited and corrected

(4)   Figures are unclear and illegible. The figure corrected and provided

(5)   References should be arranged in alphabetical order and written in a consistent manner. References reordered

Reviewer 2 Report

-The manuscript title is relevant, and informative but requested to be revised.

-The authors are requested to revise and rewrite the abstract, still missing how they accumulated the data and information, and what is the value of this study.

-The list of references is short, need to add more references.

-in the introduction part need to add proper citations.

-not define what is water infrastructure, and water assets.

-in the introduction section need to discuss climate-resilient water infrastructure and water assets.

-Under the barriers, the author can add, ensure low cost/price water instrument

-The author request to think, about whether it would be a barrier or not, to ensure climate-resilient water infrastructure, and water asset. Because many parts of the world, especially in the coastal areas, it is quite difficult to ensure the longevity of the water infrastructure, and water assets., that’s why the water investor is often scared of their finance.

-Figure 1, the text can’t be read clearly, need to reproduce the figure.

-Figure 2; under SGDs, Goal 3: Energy, needs to be revised; I think the SDGs Goal3: Good Health and Well-being.

-Requested to the author, if it was possible we can add the SDGs goals 5: Gender Equality; and Goal 11 and 12.

-the authors are requested to add “construction’ under Purpose of use of Acceptable Revenue.

-also request to add “climate resilient water infrastructure, water asset” under Paris Agreement.

Author Response

Thanks for the reviewer and I much appreciated your valuable comments to improve the paper's contents and presentations.

Comments and Suggestions for Authors

  • The manuscript title is relevant, and informative but requested to be revised. The title was revisited and changed to Water Asset Transition through Treating Water as a New Asset Class for Paradigm Shift for Water-Climate Resilient
  • The authors are requested to revise and rewrite the abstract, still missing how they accumulated the data and information, and what is the value of this study. The abstract is revisited and additions provided
  • The list of references is short, need to add more references. Two references added and please note this is a novel financial concept for water infrastructure
  • in the introduction part need to add proper citations. Citation revisited
  • -not define what is water infrastructure, and water assets. Water asset in the paper concept provided in page 3
  • -in the introduction section need to discuss climate-resilient water infrastructure and water assets. Water infrastructure discussion provided under intro
  • Under the barriers, the author can add, ensure low cost/price water instrument, the information provided as suggested
  • The author request to think, about whether it would be a barrier or not, to ensure climate-resilient water infrastructure, and water asset. Because many parts of the world, especially in the coastal areas, it is quite difficult to ensure the longevity of the water infrastructure, and water assets., that’s why the water investor is often scared of their finance.
  • Figure 1, the text can’t be read clearly, need to reproduce the figure. Text corrected and updated
  • Figure 2; under SGDs, Goal 3: Energy, needs to be revised; I think the SDGs Goal3: Good Health and Well-being. Revisited and corrected
  • Requested to the author, if it was possible we can add the SDGs goals 5: Gender Equality; and Goal 11 and 12. Thanks for the suggestions and the figure is revisited and corrected
  • the authors are requested to add “construction’ under Purpose of use of Acceptable Revenue. It is implied under the purpose of use
  • also request to add “climate resilient water infrastructure, water asset” under Paris Agreement. Information provided 

Reviewer 3 Report

This study proposes an innovative financial model for water reuse and sanitation, as a new asset class, that addresses financial market barriers and then discusses the key actions to overcome these barriers. This study introduces a very interesting approach to combat water scarcity in the future. 

My only suggestion is the paper to go under an English reviewer/editor. Other than that, the paper is novel enough to be recommended for this journal.

Author Response

Thanks for the reviewer and much appreciated your valuable comments. All comments considered along with other reviews comments

Round 2

Reviewer 1 Report

The structure of the paper has been improved and is easier to read. The corrections noted have been appropriately revised. However, some minor corrections are still needed.

(1)   The font type and size of the submitted author_response.docx and the PDF manuscript provided by the publisher differ in some areas. Authors should also check the PDF manuscript and correct any unintentional changes in font type or size. For example, there are parts in the abstract where the font size is different. The same is true on page 4.

(2)   The line spacing on page 4, lines 7-9 of the PDF manuscript is different from other parts of the manuscript.

(3)   On page 5 of the PDF document, the beginning of texts should be capitalized, full stops should be added at the end of lines, and other parts of the document should be checked for consistency.

(4)   The font type in the reference list is different and the date of search should be added to the URL.

Author Response

Comments and Suggestions for Authors

The structure of the paper has been improved and is easier to read. The corrections noted have been appropriately revised. However, some minor corrections are still needed. Thanks to the reviewer for the valuable comments to improve the paper.

 The font type and size of the submitted author_response.docx and the PDF manuscript provided by the publisher differ in some areas. Authors should also check the PDF manuscript and correct any unintentional changes in font type or size. For example, there are parts in the abstract where the font size is different. The same is true on page 4.

The font size and type checked along the paper for Calibre Body 11 size.

(2)   The line spacing on page 4, lines 7-9 of the PDF manuscript is different from other parts of the manuscript.

 The whole paper checked for single spacing and 0 and 6 before and after lines.

(3)   On page 5 of the PDF document, the beginning of texts should be capitalized, full stops should be added at the end of lines, and other parts of the document should be checked for consistency.

 The whole paper checked for capital and full stops and consistency.

(4)   The font type in the reference list is different and the date of search should be added to the URL.

The reference section uses the same font calibrs 11 size and date of research provided for URL.

Reviewer 2 Report

-the abstract is revised but needs to be concise; still missing data and information accumulation process.

-the author may include the more recent and relevant reference

-the introduction part needs citation of the fact and statement.

-in the introduction section need to discuss climate-resilient water infrastructure and water assets.

-The author request to think, about whether it would be a barrier or not, to ensure climate-resilient water infrastructure, and water asset. Because many parts of the world, especially in the coastal areas, it is quite difficult to ensure the longevity of the water infrastructure, and water assets., that’s why the water investor is often scared of their finance.

-Figure 1, the text can’t be read clearly, need to reproduce the figure.

Author Response

Comments and Suggestions for Authors

-the abstract is revised but needs to be concise; still missing data and information accumulation process.

Reference for data and information provided, see first para in the abstract

-the author may include the more recent and relevant reference . The recent reference in relation to the initiative has been provided (the GCF water sector guide)

-the introduction part needs citation of the fact and statement. All reported facts are referenced as appropriate and other facts are GCF analysis.

-in the introduction section need to discuss climate-resilient water infrastructure and water assets. Please see the third para in the introduction, where water infrastructure discussed

-The author request to think, about whether it would be a barrier or not, to ensure climate-resilient water infrastructure, and water asset. Because many parts of the world, especially in the coastal areas, it is quite difficult to ensure the longevity of the water infrastructure, and water assets., that’s why the water investor is often scared of their finance.

The author is in agreement with the reviewer, and therefore this paper present the new asset class, that would ensure revenue streams that support operational and maintenance cost. If the reviewer means the longevity against extreme events such as sea level rise and flood inundation, this should be taken into account in the design phase including IFD analysis and selection of location, etc.

-Figure 1, the text can’t be read clearly, need to reproduce the figure.

The original figure provided for printing and publishing
